# An Orthotopic Model of Glioblastoma Is Resistant to Radiodynamic Therapy with 5-AminoLevulinic Acid

**DOI:** 10.3390/cancers14174244

**Published:** 2022-08-31

**Authors:** Charles Dupin, Jade Sutter, Samuel Amintas, Marie-Alix Derieppe, Magalie Lalanne, Soule Coulibaly, Joris Guyon, Thomas Daubon, Julian Boutin, Jean-Marc Blouin, Emmanuel Richard, François Moreau-Gaudry, Aurélie Bedel, Véronique Vendrely, Sandrine Dabernat

**Affiliations:** 1BRIC (BoRdeaux Institute of onCology), UMR1312, INSERM, University of Bordeaux, F-33000 Bordeaux, France; 2Radiotherapy Department, CHU Bordeaux, F-33000 Bordeaux, France; 3Tumor Biology and Tumor Bank Laboratory, Bordeaux University Hospital, F-33000 Bordeaux, France; 4Animalerie Mutualisée, Service Commun des Animaleries, University of Bordeaux, F-33000 Bordeaux, France; 5INSERM, BPH, U1219, University of Bordeaux, F-33000 Bordeaux, France; 6Service de Pharmacologie Médicale, Bordeaux University Hospital, F-33000 Bordeaux, France; 7CNRS, IBGC, UMR5095, University of Bordeaux, F-33000 Bordeaux, France; 8Biochemistry Department, Bordeaux University Hospital, F-33000 Bordeaux, France

**Keywords:** glioblastoma, radiotherapy, 5-AminoLevulinic Acid (5-ALA), protoporphyrin IX, radiodynamic therapy

## Abstract

**Simple Summary:**

Radiosensitization of glioblastoma is a major ambition to increase the survival of this incurable cancer. Before surgery, oral administration of 5-aminolevulinic acid leads to the accumulation of fluorescent protoporphyrin IX, preferentially in the cancer cells rather than in the normal brain. This property is used to optimize the resection of the tumor under fluorescent light. Protoporphyrin IX may also carry radiosensitization activity. To test this hypothesis, we designed a robust murine preclinical model of glioblastoma with tumors implanted into the brain of mice treated by fractionated radiotherapy, as for humans. Despite the specific accumulation of porphyrins in glioblastoma, there was no radiosensitization. We confirmed these results in in vitro 3D patient-derived spheroids. Radiosensitization by molecules such as porphyrins needs more exploration for application in glioblastoma treatment.

**Abstract:**

Radiosensitization of glioblastoma is a major ambition to increase the survival of this incurable cancer. The 5-aminolevulinic acid (5-ALA) is metabolized by the heme biosynthesis pathway. 5-ALA overload leads to the accumulation of the intermediate fluorescent metabolite protoporphyrin IX (PpIX) with a radiosensitization potential, never tested in a relevant model of glioblastoma. We used a patient-derived tumor cell line grafted orthotopically to create a brain tumor model. We evaluated tumor growth and tumor burden after different regimens of encephalic multifractionated radiation therapy with or without 5-ALA. A fractionation scheme of 5 × 2 Gy three times a week resulted in intermediate survival [48–62 days] compared to 0 Gy (15–24 days), 3 × 2 Gy (41–47 days) and, 5 × 3 Gy (73–83 days). Survival was correlated to tumor growth. Tumor growth and survival were similar after 5 × 2 Gy irradiations, regardless of 5-ALA treatment (RT group (53–67 days), RT+5-ALA group (40–74 days), HR = 1.57, *p* = 0.24). Spheroid growth and survival were diminished by radiotherapy in vitro, unchanged by 5-ALA pre-treatment, confirming the in vivo results. The analysis of two additional stem-like patient-derived cell lines confirmed the absence of radiosensitization by 5-ALA. Our study shows for the first time that in a preclinical tumor model relevant to human glioblastoma, treated as in clinical routine, 5-ALA administration, although leading to important accumulation of PpIX, does not potentiate radiotherapy.

## 1. Introduction

Glioblastoma (GM) is the most common primary malignant brain tumor. Since 2005 and temozolomide concomitant treatment and adjuvant to radiotherapy (RT), no new drug has shown its superiority as a first-line treatment [1]. Many drugs and strategies have been evaluated including anti-angiogenic agents [2,3], the EGFR–tyrosine kinase inhibitor erlotinib [4], anti-PD1 immunotherapy [5], and vaccination of EGFRvIII glioblastoma [6]. Radiosensitizing agents need to reach high concentrations in the tumor tissue, but the blood-brain barrier limits the tumor-to-tissue ratio, particularly in the Magnetic Resonance Imaging non-enhanced zone [7]. Thus, novel radiosensitization agents are still needed and should have the property to go across the blood-brain barrier.

The ubiquitous heme metabolic pathway results in heme synthesis and cytochrome production [8]. The 5-Aminolevulinic Acid (5-ALA) is a heme precursor. The exogenous administration of 5-ALA leads to protoporphyrin IX (PpIX) accumulation because of the saturable activity of the ferrochelatase, the last enzyme of the pathway. PpIX emits fluorescence at 625 nm after excitation at 405 nm. A specific accumulation of fluorescent porphyrins occurs in glioblastoma multiform compared to the normal brain [9]. This property was exploited in a subsequent randomized phase III clinical trial to compare classical neurosurgery to fluorescence-guided surgery after 5-ALA administration. PpIX-guided tumor visualization increased complete resection from 36% to 65%. Progression-free survival was better, however, at the cost of neurological deficits with fluorescence-guided surgery, and this strategy did not improve overall survival [10]. In the same way, a recent review of fluorescence-guided surgery showed an increase in the overall survival and progression-free survival compared to the white light group [11]. Post-operative neurologic outcomes were better, the same, and worse after fluorescence-guided surgery in 42%, 35% and 23%, respectively.

PpIX accumulation after 5-ALA administration is now tested as a photosensitizing strategy in glioblastoma [12,13]. Although PpIX photosensitization of tumor cells is efficient, interstitial phototherapy used to improve in-depth light irradiation could be toxic for healthy surrounding tissue.

Porphyrins may potentialize radiotherapy by radiodynamic therapy [14,15,16]. If PpIX accumulates specifically in vivo in human glioblastoma, 5-ALA administration could be combined with spatially targeted radiotherapy. Porphyrin accumulation-based radiodynamic therapy showed encouraging results in a head and neck cancer model with the invalidation of Uroporphyrinogen III decarboxylase UROD, the fifth enzyme of the pathway [17]. Moreover, potentiation of radiotherapy was obtained with photofrin II or a hematoporphyrin derivative in a hepatic tumor model [18] and with 5-ALA-induced PpIX accumulation in a melanoma brain metastasis model [14]. In subcutaneous glioblastoma models, 5-ALA treatment, resulting in PpIX accumulation, radiosensitized the U87 and U251 cell lines [14,19,20] and the 9 L gliosarcoma cells [21].

Porphyrin-based radiodynamic therapy for glioblastoma is still in need of efficacy demonstration in a relevant orthotopic xenograft model, with a radiotherapy treatment sequence paralleling that of humans.

The first objective of this work was to establish a robust preclinical patient-derived xenograft (PDX) model, treated with 5-ALA, and irradiated according to a scheme paralleling human routine radiation therapy. Next, we assessed the radiosensitization potential of PpIX after 5-ALA in vitro treatment of 3D spheroids derived from patients’ tumors.

## 2. Materials and Methods

### 2.1. Cell Culture 

The P3 cell line is a patient-derived GM stem cell line from Patient 3 published by Sakariassen et al. [22] and were well characterized by other studies [23,24,25,26]. GG6 and GG16 were obtained by Joseph et al. [27] from surgical samples pathologically confirmed as GB and characterized further [26]. GG16 was a giant cell glioblastoma. These 3 cell lines were derived from IDH wild-type GB patients.

P3 cells were transduced by the lentiviral virus coding for Luciferase [28]. Cells were cultured in Neurobasal medium Gibco, with B27 supplementation, Penicillin, streptomycin, sodic heparin, and 20 ng/mL FGFbeta. For the spheroid condition, a constant concentration of 17 µL/mL medium of Accutase StemPro^®^ Gibco was used to avoid cell adhesion to the dishes.

### 2.2. Spectroscopic Dosage

First, protein extraction was performed using Pierce BCA Protein Assay Kit (ThermoFisher, Rockford, IN, USA). The spectroscopic measures were carried out on a HITACHI F-4500 fluorescence spectrophotometer using a 404 nm excitation laser, and an emission spectrum from 550 nm to 700 nm. Each measure was reported to protein concentration. Calibration was set according to a known porphyrin calibrant by ChromSystems^®^.

### 2.3. Porphyrin Concentrations Were Measured by Cytometry

Accury^®^ C6Plus Flow Cytometer (Brussels, Belgium) was used. To estimate porphyrin accumulation, mean fluorescence for all cells was quantified, after excitation at 488 nm and signal recovering at 670 LP.

### 2.4. Spheroid Growth and Viability

To measure the spheroid growth, a brightfield picture was taken for each spheroid, and the area was determined using thresholding with ImageJ software (version 1.53c, Bethesda, MD, USA), as previously described by Guyon et al. [25]. To explore cell viability in spheroids, 4 spheroids were dissociated and incubated for 20 min in PI Propidium Iodide (PI)/Calcein staining dead and alive cells, respectively. Stained cells were analyzed by cytometry. For qualitative evaluation of live/dead cell status in spheroids themselves, whole spheroids were incubated with PI/Calcein for 20 min and imaged in a fluorescence microscope (Nikon Eclipse Ti-U). Spheroid formation assays with GG6 and GG16 resulted in spheroids of various sizes, probably linked to cell adhesion properties. To limit intra-experiment variations, individual spheroids were formed, then exposed to 5-ALA and/or radiotherapy. Their growth was individually followed after treatment.

### 2.5. Cell Culture for Radiotherapy

To determine the anti-tumor activity of RT combined with 5-ALA, 400,000 dissociated cells were cultured in T25 flasks for 2 days before adding 5-ALA to reach 0.3 mM to 1 mM final concentrations. Flasks were incubated for 4 h in the dark, before irradiation. Radiotherapy was carried out on XENX Xstrahl^®^ irradiator, 220 kV, 1 Gy/min, at 0–4–10 Gy. After radiotherapy, spheroids were transferred in 15 mL tubes, centrifuged (2000× *g* for 10 min), washed with PBS, centrifuged again, incubated with accutase^®^ for 12 min, counted in Kova cells, resuspended in NB medium and distributed in 100 µL-10,000 cells/well in low binding U-bottom 96 well plates.

All manipulations were done with minimal light exposure after 5-ALA incubation, particularly during the first 24 h.

### 2.6. Mouse Model

Immunodeficient RAGγ2C^−/−^ [29] were housed and treated in the animal facility of Bordeaux University (“Animalerie Mutualisée Bordeaux”). All experiments were approved by the “Ministère de l’Enseignement Supérieur, de la Recherche et de l’Innovation (MESRI)” (authorization number 2018021416296098), and were carried out in accordance with the approved protocols.

Five spheroids of 10^4^ cells were stereotactically implanted into the right cerebral cortex of each mouse, 2.2 mm left of the bregma at a 3.3 mm depth, using a Hamilton syringe fitted with a needle (Hamilton, Bonaduz, Switzerland). Tumor growth was monitored by bioluminescence on Biospace Lab^®^ after intraperitoneal injection of 150 mg/kg de D-Luc (Promega, E264X) every week on Thursdays.

### 2.7. Radiotherapy

Radiotherapy was performed under general anaesthesia by isoflurane 2.5% at 220 kV on XENX Xtrahl^®^. The dose was controlled by a habilitated radiation physicist with EBT3 Gafchromic films (Ashland).

Before each mouse irradiation, position control was performed with a portal image to ensure radiotherapy accuracy. Treatment was done with 2 opposite lateral rectangular fields (15 × 20 mm) at 2.55 Gy/minute. Radiotherapy was performed 3 times a week (Monday-Wednesday-Friday)

Three radiotherapy regimens were evaluated 3 × 2 Gy, 5 × 2 Gy and 5 × 3 Gy, 3 times a week 43 days after orthotopic xenografts, and compared to an untreated group (n = 5/group).

To test the radiosensitization potential of porphyrin accumulation, 3 groups of 11 mice were used. One group received only 5-ALA, one group received radiotherapy alone and the third group received both. We treated the mice on day 39.

5-ALA injection was carried out at a final dose of 100 mg/kg with an intraperitoneal injection of approximately 100 µL, according to the mouse weight. The best window for 5-ALA injection during early tumor growth was determined to obtain a maximum PpIX concentration in tumors and a minimum concentration in healthy tissues, during irradiation.

For the pharmacokinetic experiment, after mouse sacrifice, the brain was collected. Parts of the tumor and the normal frontal brain were dissected. After tissue crushing, the same procedure was applied as for the spheroids, and porphyrin concentrations were spectroscopically determined and normalized to protein concentrations.

### 2.8. Statistical Analyses

Spheroid sizes and bioluminescence values were compared by ANOVA. Survivals were assessed by log rank tests. To determine the needed sample size to detect a difference between radiotherapy and radiotherapy+5-ALA, we used the results of the pilot study with different radiotherapy regimens (Figure 1). We hypothesized that 5-ALA administration with a 10 Gy (5 × 2 Gy) dose of radiotherapy could decrease the tumor growth as a total dose of 15 Gy (5 × 3 Gy, without 5-ALA) would do. The G*Power 3.1 software (Heinrich-Heine-Universität Düsseldorf, Düsseldorf, Germany; http://www.gpower.hhu.de/, accessed on 15 December 2020) was used to calculate the appropriate minimum number of animals with an alpha-error type of 5% and a power of 95%. The minimum sample size was 5 mice per group [30]. To assess differences in tumor growth with a power of 99% and an alpha-type error of 2% 11 mice were needed per group.

## 3. Results

### 3.1. Orthotopic PDX Treated by Radiotherapy

To test the relevance of 5-ALA treatment in radiodynamic therapy, we needed to develop an orthotopic PDX, treated with whole-brain irradiation (Figure 1A). For this, we used the P3 cells [25]. A unique 10 Gy dose on P3 cell-xenografted mice led to a long survival, suggesting that P3 cells were radiosensitive. To parallel the clinical situation, delivering fractionated doses between 2 and 3 Gy, and keeping in mind that we needed non-curative radiotherapy, we tested three fractionated radiation regimens with potential low and high efficiencies on tumors: 6 Gy (3 × 2 Gy, three times a week), 10 Gy (5 × 2 Gy, three times a week) and 15 Gy (5 × 3 Gy, three times a week). Luminescent P3 cell-derived tumor growth was followed by live bioimaging (Figure 1B). Groups were formed to obtain similar tumor bioluminescence averages 34 days post-implantation, before irradiation initiation, at day 39. The control untreated tumors displayed exponential bioluminescence. All treated groups showed a breakpoint in tumor growth from day 55 compared to the control group, and we observed various delays to regrowth, according to the treatment regimen (Figure 1C). Expectedly, the 6 Gy-treated group had an increase in bioluminescence at day 69 while growth curves of 10 Gy and 15 Gy regimens were statistically different after 76 days (Figure 1C). Thus, irradiation had a suspensive effect, and the time of regrowth depended on the delivered dose.

Survival was statistically different between all groups with clear dose-dependent mortality (Figure 1D) Median survivals were 59 days, 84 days, 96 days, and 119 days for control, 6 Gy, 10 Gy, and 15 Gy groups, respectively. Only one mouse of the 10 Gy group died during the survival span of the 6 Gy group. Overall, the groups displayed strong homogeneity in survival with death occurring in non-overlapping periods. Thus, mice outcomes strictly paralleled tumor regrowth.

In conclusion, to maximize the impact of 5-ALA treatments in combination with fractionated radiotherapy, the regimen of 10 Gy appears the best as it exposed tumor cells five times to the treatment. In addition, it delivered radiotherapy at 2 Gy/fraction like in the Stupp et al. protocol, routinely used in clinics [31].

### 3.2. Defining the Time to Inject 5-ALA before Irradiation

There is a delay between 5-ALA treatment and PpIX accumulation in tumor cells [10,32,33]. To determine the best window of irradiation post-5-ALA injection, 5-ALA was injected intraperitoneally and mouse brains were collected at different time points to quantify the porphyrin content of each tumor and its normal surrounding tissues. Porphyrins accumulated preferentially in tumors compared to the normal brain as soon as 2 h after 5-ALA injection (Figure 2). Concentrations tripled after 4 h. Although normal brain tissue also accumulated porphyrins after 4 h, concentrations were half of that found in the tumor tissue. We chose to inject 5-ALA 4 h before irradiation, considering that intra-tumor porphyrin concentration would be at their best while healthy tissues would be less impacted because of the lower accumulation. Interestingly, the only residual porphyrin concentrations were found in healthy cells after 24 h (Figure 2), suggesting that in a multifractionation radiation schedule, the previous injection of 5-ALA would not contribute to intracellular porphyrin content and potential radiosensitization activity.

### 3.3. 5-ALA Does Not Radiosensitize the P3 Orthotopic PDX Model Treated by the Whole Brain Fractionated Radiotherapy

As previously mentioned, to increase chances to observe radiosensitization, we used the irradiation protocol that exposed tumors five times to the combination treatment 5-ALA/radiotherapy.

To test the radiosensitization potential of porphyrin accumulation, three groups of 11 mice were used. One group received only 5-ALA, one group received radiotherapy alone and the third group received both. We treated the mice on day 39. Median survivals for 5-ALA, radiotherapy only and 5-ALA + radiotherapy were 56 days, 96 days, and 102 days, respectively. Overall tumor growth was very similar to that observed when testing the different radiotherapy regimens, with a split of the tumor growth curves at around day 50, and exponential growth of the tumors treated with 5-ALA only (Figure 3A), leading to the death of all the mice by day 60 in this group (Figure 3B). This suggests that 5-ALA alone does not affect tumor growth. Radiotherapy impacted the tumor growth as previously observed since tumors showed stable luminescence signals up to 83 days after treatment, then relapsed and killed all the mice around 100 days post-transplantation. This highlights the high comparability between independent experiments, showing the robustness of the model. Expectedly, there was a significant difference in survival between the 5-ALA only and the radiotherapy only groups (HR = 3.52[1.29–9.63] *p* < 0.0001), or between the 5-ALA only and the Radiotherapy + 5-ALA groups (HR = 3.52[1.29–9.63] *p* < 0.0001 for 5-ALA vs. 5-ALA+RT) (Figure 3C).

When 5-ALA administration was included in the radiotherapy regimen no difference was observed, neither on the tumor growth kinetics (Figure 3A) nor on the mouse survival (Figure 3B), as compared to radiotherapy only (HR = 1.57[0.67–3.69], *p* = 0.24). This suggests that in our settings 5-ALA administration failed to radiosensitize P3 cell-derived xenografts.

### 3.4. 5-ALA Treatment Radiosensitization Evaluation on P3 Cell-Derived Spheroids

The absence of radiosensitization was puzzling, especially with regard to published data, using 5-ALA [14,32,33]. To test whether radiosensitization would be more easily achievable in vitro, we determined the growth of P3 cell-derived spheroids by measuring spheroid areas, after a single dose of irradiation, with or without 5-ALA pre-treatment. We also quantified the proportions of live and dead cells within each spheroid after different treatments.

Untreated spheroids showed linear growth (Figure 4A). P3 cell-derived spheroids were radiosensitive in a dose-dependent manner, confirming the observed in vivo efficacy of radiotherapy. Of note, even at a high dose (10 Gy), spheroids did not die but displayed growth arrest. There was no dose-effect of 5-ALA treatment in the absence of radiotherapy (Figure 4B). In the same way, increasing doses of 5-ALA did not alter the P3 cell-derived spheroid growth, regardless of the irradiation dose (Figure 4C). These results were confirmed by the evaluation of the proportions of live/dead cells of the spheroids. The proportions were very similar for a given irradiation dose, with or without 5-ALA treatment (Figure 4D,E). Together, these results confirm that P3 cell-derived spheroids were radiosensitive but were not radiosensitized by 5-ALA pre-treatment.

As we confirmed that porphyrins accumulated in tumor tissues after 5-ALA administration, we verified that this was also true in the P3 cell-derived spheroids. We found that porphyrin concentrations in spheroid extracts, measured by spectroscopy, was proportional to the mean fluorescence intensities in 5-ALA treated cells, as determined by flow cytometry (R^2^ = 0.9984, *p* < 0.0001) (Figure 5A). We also showed 24 h after a 4-h 5-ALA exposure, there was important porphyrin accumulation in P3 spheroids, at the time of radiotherapy treatment. As in vivo, cells had cleared porphyrins 24 h after 5-ALA exposure (Figure 5B).

Taken together, this set of data confirms that despite porphyrin accumulation, to higher levels than what was obtained in vivo, 5-ALA treatment did not radiosensitize P3 cell-derived spheroids.

As these results suggest that 5-ALA exposure did not radiosensitize P3 cell-derived spheroids, we tested the GG6 and the GG16 cells, two additional patient-derived spheroid-forming GB cell lines. First, we confirmed that under 5-ALA treatment, they accumulated porphyrins, as evidenced by flow cytometry (Figure 5B). As for P3 spheroids, untreated cells showed very little fluorescence signals. 5-ALA induced the appearance of highly fluorescent fluorocytes. As for P3 spheroids, porphyrins were cleared after 24 h.

Next, we evaluated the radiosensitization property of 5-ALA treatment in GG16 and GG6 spheroids. Both cell lines were radiosensitive, in a dose-response manner, as evidenced by the reduced spheroid growth under radiotherapy (Figure 6A,B for 0 Gy, Figure 6C,D for 4 Gy and Figure 6E,F for 10 Gy, black curves).

The GG16 spheroids were not radiosensitized in the presence of 5-ALA, as evidenced by the overlapping green (+ 5-ALA) and black (no 5-ALA) spheroid growth curves (Figure 6A,C,E). The GG6 spheroid growths were similar at 0 Gy and 4 Gy (pink curves, Figure 6B–D), but 5-ALA treatment affected their growth at 10 Gy (pink curve, Figure 6F). GG16 spheroids were stably recovered over time, regardless of 5-ALA treatment (Figure 6G). By contrast, we observed frequent GG6 spheroid dissociation in the presence of 5-ALA, with numerous floating dead cells. Indeed, only 50% of the untreated or 4 Gy-treated spheroids were recovered at 7 and 14 days under 5-ALA treatment (Figure 6H, bold and dashed pink curves), whereas at least 85% of the spheroids were still growing without 5-ALA (Figure 6H bold and dashed black curves). At 10 Gy, 47% of the spheroids showed dissociation at 14 days, while all the 5-ALA treated spheroids had disappeared (Figure 6H, dotted black and pink lines, respectively). These results suggest that 5-ALA treatment was toxic for the GG6 cells, but did not increase the radiation toxicity.

All together, these results show that as for the P3 cell line, 5-ALA did not radiosensitize the GG6 and the GG16 cell lines in vitro making in vivo evaluation useless.

## 4. Discussion

To our knowledge, this study is the first to evaluate 5-ALA-induced radiosensitization potential using whole-brain radiotherapy in an orthotopic glioblastoma PDX.

In vitro models are of critical importance for translational medicine. The 3D spheroid model progressively appeared as the gold standard of preclinical studies in glioblastoma. Tumor-derived GB cells cultured in 3D displayed radiosensitization with temozolomide and bevacizumab, while erlotinib was inefficient. Interestingly, in vitro sensitization profiles were similar to patients’ tumor responses [34]. The same cells cultured as monolayers showed contrasting results since temozolomide radiosensitized only one patient-derived GB line, bevacizumab had no effect, and erlotinib did radiosensitize the cells. Previous studies testing 5-ALA potential first tested cell response in 2D cultures, then moved to a further demonstration in vivo, potentially introducing a selection bias. Moreover, subcutaneously implanted tumors, that were widely used in studies testing 5-ALA, may not perfectly reflect the tumor in its microenvironment. Indeed, tumor vasculature depends on the implantation site, as well as the recruitment of supportive stromal cells. Together, these differences may impact tumor response [35]. Our results confirmed the reproducibility of tumor cell response whether they were evaluated as spheroids in vitro or orthotopically grafted in vivo.

Using a clinically relevant model, we found that 5-ALA administration did induce porphyrin accumulation in tumor cells, but did not radiosensitize them. Importantly, we validated the lack of effect in vitro on three independent 3D human spheroid models. In a subcutaneous syngeneic rat model of gliosarcoma, local fractionated radiotherapy (10 Gy, 2 Gy/day for 5 days) diminished tumor growth. This effect was enhanced with 5-ALA administration (100 mg/kg in tail vein 3 h before irradiation). In this study, where rats were housed under a 12 h light/dark cycle, 5-ALA could have an effect by itself, as suggested by the 5-ALA treatment tumor growth curve that was below the untreated curve. A phototherapeutic effect in subcutaneous tumors cannot be excluded [21]. In addition, this immune-competent model is interesting because the authors found increased macrophage tumor infusion in the 5-ALA-treated rats probably contributing to the tumoricidal effect. Subcutaneous implantations of glioblastoma U87MG and U251MG cell lines treated with fractionated radiotherapy (whole-brain, 60 Gy, 2 Gy/day, 5 days a week), and 5-ALA (60–120 mg/kg per os 4 h before irradiation) failed to find a radiosensitization effect in U87MG-transplanted mice at the endpoint of the experiment, by contrast to U251MG-transplanted mice [14]. This suggests heterogeneity in in vivo tumor cell response to 5-ALA-mediated radiosensitization.

Besides glioblastoma, other tumor cell types were challenged with radiotherapy and 5-ALA co-treatments. Intracranial xenografts of murine melanoma cells were treated with radiotherapy (whole-brain, 14 Gy, 2 Gy/day, 7 days), resulting in diminished tumor growth. This effect was enhanced by 5-ALA (200 mg/kg intraperitoneally 4 h before irradiation) [33]. However, the growth capacity of melanoma cells is drastically different from that of glioblastoma cells, since grafted mice died 9 days after implantation. This high rate of proliferation may identify an early effect of the combination treatment that may not persist if tumors can recover from the initial toxic crisis. In agreement with our results, subcutaneous PC-3 prostate tumors were not radiosensitized by 5-ALA (100 mg/kg intravenous). Authors noticed improved efficiency of radiotherapy (a single 4 Gy dose) only when they used carbamide peroxide delivered intratumorally [15] in combination with 5-ALA, suggesting that this model needed higher levels of reactive oxygen species.

Noticeably, the irradiation schedules were different between studies. Some were single doses [15,21,32] or multiple with at least 0.9 Gy (0.3 Gy/every day 3 times) [32] and up to 60 Gy (2 Gy/day/5 days/week/6 weeks) [14], compared to our treatment plan of 10 Gy (2 Gy/day/3 days/10 days). Radioresistance in glioblastoma depends on multiple biological processes [36] together with the tumor type (stem cells, tumor heterogeneity, metabolic alteration, DNA damage repair...) and the model itself (tumor microenvironment, hypoxia). As none of the published studies combine the same host/tumor/graft site, it is difficult to compare them. It is also not necessary to reproduce the exact same experiments to obtain interpretable results [14]. We validated our model by (i) checking the efficacy of dose-response radiotherapy, and (ii) making sure that porphyrins accumulated preferentially in the tumor. A multi-fractionated treatment was close to the human radiotherapy regimen and allowed multiple exposures to porphyrins during irradiation.

Our study presents some limits. The irradiation window covered the whole mouse brain but copied the whole brain human routine protocol by opposed pair field plan. More accurate radiation therapy can be delivered, as shown by Rutherford et al. [37] when adaptative tumor brain imaging is available. This may be challenging, though, in the multifractionation treatment of mice. Overall, our irradiation setting was as close as possible to what is routinely used in patients and was not toxic for the mice. Of note, we used luciferase/luciferin for the tumor follow-up. Luciferase photon emission could interfere with a phototherapeutic effect. However, the administration of 5-ALA and luciferin were always distant by 24 h, and we showed that PpIX was cleared during this period. Moreover, we noted that 5-ALA only-treated tumors displayed growth similar to that of untreated tumors, and survivals were the same. Illumination induced by luciferin is many orders of magnitude below the needed energy to induce a phototherapeutic effect [38], even if self-exciting photodynamic therapy is a promising field of research [39].

Heme pathway modulation deserves attention in radiosensitization research. Indeed, the mechanism of porphyrin-related radiosensitization is still unclear and may be independent of their photoactivability [17]. Moreover, even regarding photoactivability, all porphyrins might not be equivalent. This hypothesis is supported by the fact that photodermatosis in porphyria does not occur for all the enzymatic deficiencies of the heme pathway. For example, ferrochelatase deficit-accumulating free PpIX in the skin leads to photosensitivity, as well as gain of function of 5-ALA synthase 2 (ALAS2) leading to accumulation of PpIX-Zinc and photodermatosis. Alternative porphyrin accumulation may improve radiosensitization, as their observed photosensitization in porphyria. Indeed, uroporphyrinogen decarboxylase (UROD) downregulation altered iron homeostasis in a head and neck cancer model, produced reactive oxygen species, and enhanced radiotherapy [17]. UROD blockade results in the accumulation of uroporphyrinogen III. In the same way, uroporphyrinogen III synthase (UROS) deficit results in the accumulation of type I porphyrins, with the strongest impact on skin lesions [40]. These intermediates might present properties towards response to radiotherapy different from other porphyrins. This field deserves to be explored.

## 5. Conclusions

This is the first study testing the radiosensitization potential of porphyrin accumulation induced by 5-ALA in orthotopically grafted patient-derived glioblastoma cells and in 3D in vitro cultures from three patient-derived models. Using a clinically relevant fractionated radiotherapy regimen, efficient on the tumor cells, we did not observe diminished tumor growth, increased survival, or impacted spheroid formation and growth in the presence of 5-ALA. Although this result does not confirm some of the published studies in the field, the exploration of alternative porphyrin accumulation by modulating the heme pathway enzymes deserves close attention.

## Figures and Tables

**Figure 1 cancers-14-04244-f001:**
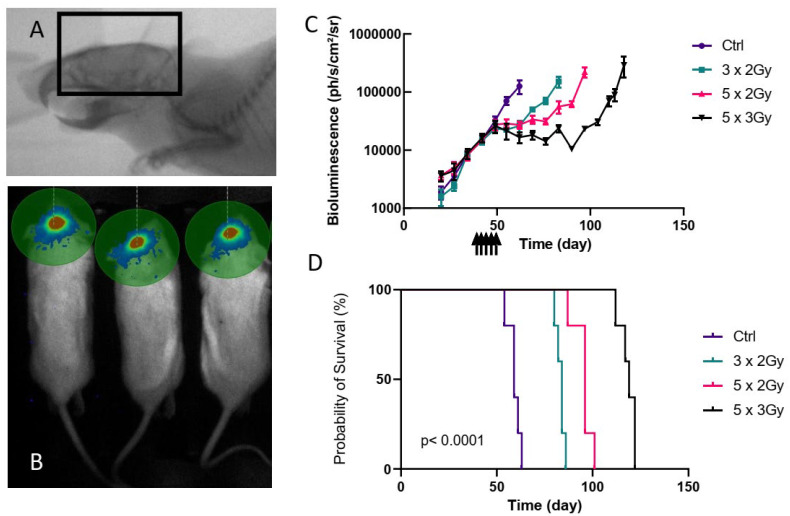
Modelling fractionated radiotherapy. (**A**) Portal imaging of mouse. The radiation field is delineated into the black rectangle. (**B**) Example of tumor bioluminescence after intraperitoneal injection of D-Luciferin showing signal limited to the brain. (**C**) Tumor follow-up by bioluminescence (Mean +/− SEM). N = 5/group. Arrows indicate the days after implantation when mice received fractionated radiotherapy (6 Gy, 10 Gy, and 15 Gy). The control group did not receive radiotherapy. (**D**) Survival probabilities (*p* < 0.0001 according to a log rank test).

**Figure 2 cancers-14-04244-f002:**
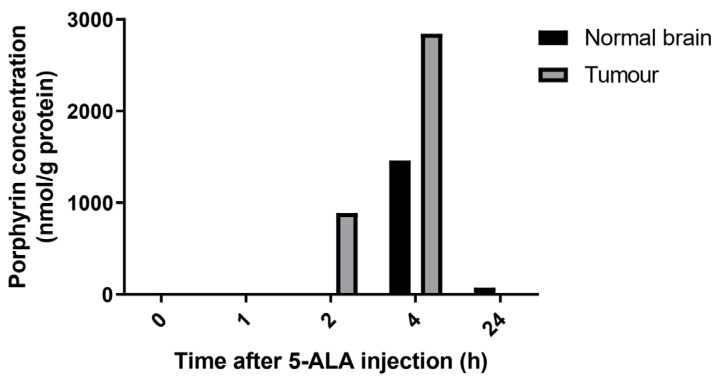
Pharmacokinetic of porphyrin concentration in normal brain, and tumor tissue according to time of 5-ALA intra-peritoneal injection (100 mg/kg).

**Figure 3 cancers-14-04244-f003:**
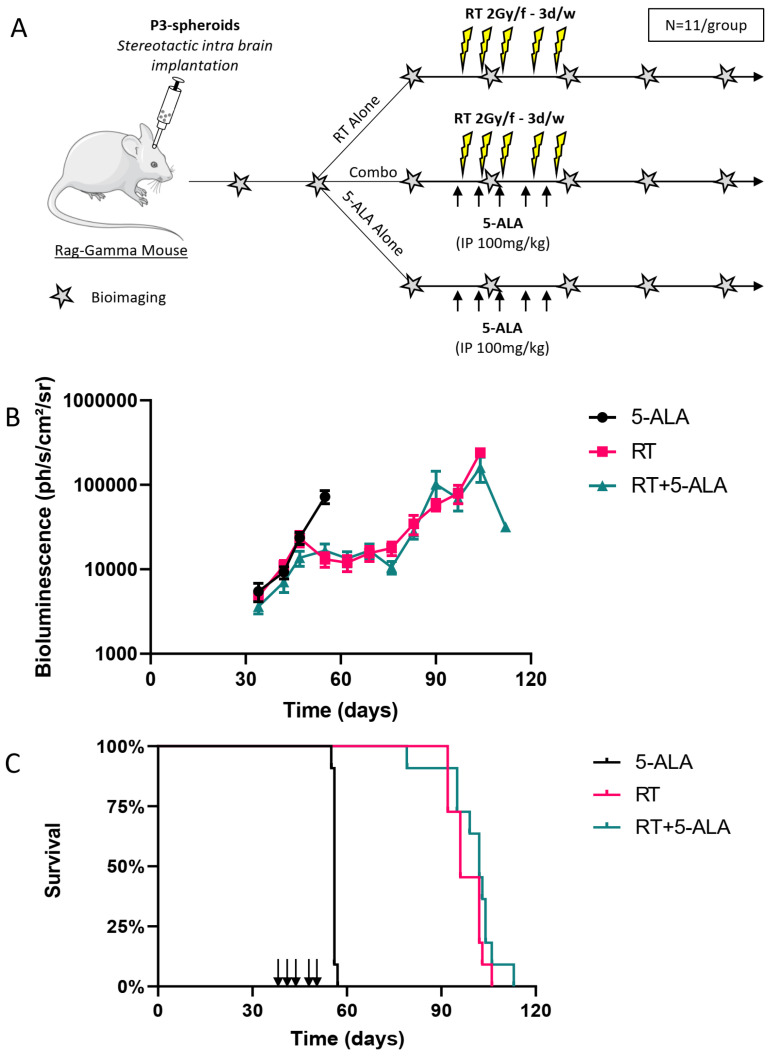
Combination treatment of 5-ALA and radiotherapy. (**A**) experimental protocol (details in materials and methods). Black arrows indicate 5-ALA administration 4 h before irradiation. Each star corresponds to luminescence imaging. (**B**) Tumor follow-up by bioluminescence (Mean +/− SEM). N = 11/group. Arrows indicate the days after implantation where mice received fractionated radiotherapy five times 2 Gy +/− 5-ALA. The control group received 5-ALA only. (**C**) Corresponding survival curves./f: per fraction; 3 d/w: 3 days a week; IP: intraperitoneal. Mouse picture provided by Servier Medical Art.

**Figure 4 cancers-14-04244-f004:**
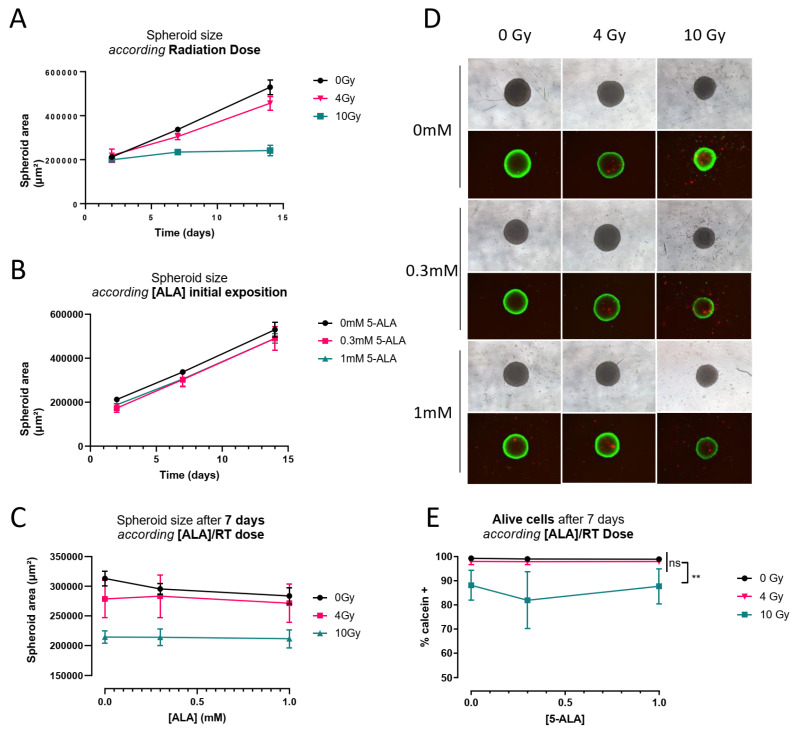
Irradiation and 5-ALA impact on spheroid growth (**A**) Spheroid size according to radiation dose without 5-ALA. (N = 3 experiments with three spheroids per condition in each experiment) (**B**) Spheroid size according to [5-ALA] initial exposure (**C**) Spheroid size according to [5-ALA] and radiation dose on day 7. (**D**) Spheroid in white light after PI/Calcein (red/green) staining. (**E**) Cytometric quantification of live cells in spheroid by calcein fixation and cytometric quantification, after 7 days (four spheroids by condition, three independent experiments). There are significantly fewer live cells compared to 0 Gy and 4 Gy. ** *p* < 0.005, ns = non significant, ANOVA test.

**Figure 5 cancers-14-04244-f005:**
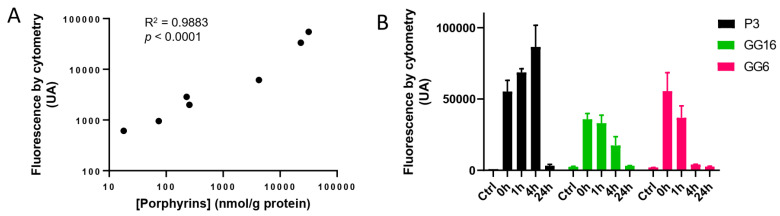
Porphyrin accumulation after 5-ALA exposition. (**A**) Porphyrin concentration determined by spectroscopy correlates fluorocyte mean fluorescence intensity in P3 cells as determined by flow cytometry. R^2^ = 0.9883, *p* < 0.0001, [Y = 1.59x + 867.2] (**B**) Fluorescence before incubation (Ctrl) or after incubation of P3 cells incubated with [5-ALA] = 1 mM during 4 h (N = 3 for P3, GG16 and GG6).

**Figure 6 cancers-14-04244-f006:**
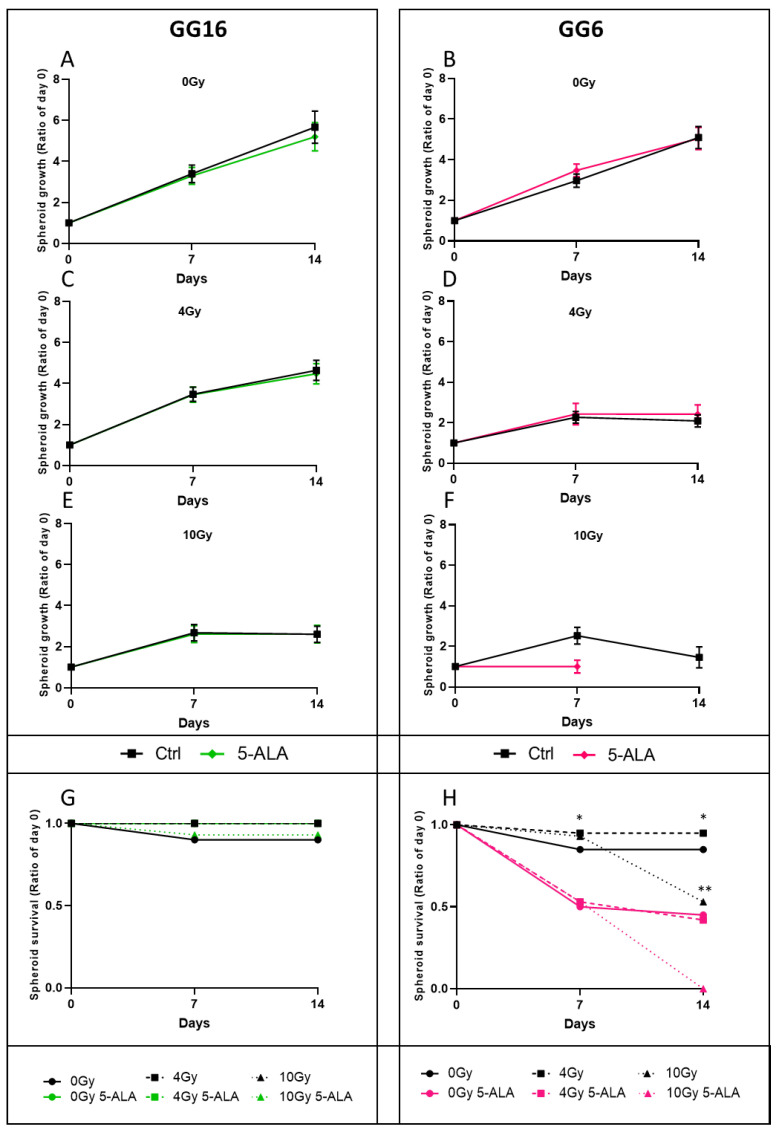
GG16 and GG6 spheroid growth and survival under radiotherapy and 5-ALA exposure. Individual spheroids obtained from the GG16 (left column) and GG6 (right column) cell lines were irradiated or not with or without 5-ALA exposure (green for GG16 with 5-ALA and pink for GG6 with 5-ALA and black curves for corresponding control), and were individually followed over a period of 14 days (n = 15 to 20 spheroids per condition), for growth (**A**–**F**) and survival (**G**,**H**). * *p* < 0.05 0/4/10 Gy condition without 5-ALA vs. with 5-ALA. ** *p* < 0.05 Ctrl/5-ALA condition 0 Gy or 4 Gy vs. 10 Gy.

## Data Availability

Not applicable.

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
