# Peer review of "An Orthotopic Model of Glioblastoma Is Resistant to Radiodynamic Therapy with 5-AminoLevulinic Acid"

_cancers, 2022, doi:10.3390/cancers14174244_

Round 1

Reviewer 1 Report

The authors investigated radiodynamic effect of 5-aminlevulinic acid (ALA) in brain tumor model which inoculated P3 cell (human stem-like glioblastoma) into the brain of immunodeficient mice. Although 5-ALA can accumulate much amount of Protoporphyrin IX(PpIX) in P3-derived spheroids, radiodynamic effect of 5-ALA was not observed in both P3-derived spheroids and orthotopic model. This study is well designed, and interesting. However, some questions should be raised.

Major concern

1)Radiosensitizing effect of porphyrin compounds have been observed in cancer cells. Because 5-ALA is prodrug of heme, amount of 5-ALA-induced porphyrin compounds (i.e. PpIX) is low in cancer cells, compared to other porphyrin, such as HpD and Photofrin. Therefore, radiodynamic effect of 5-ALA is considered as low in cancer cells via the single dose radiation. However, acuity of porphyrin synthesis has changed in cancer cells after radiotherapy with some variation. If you confirm accumulation of 5-ALA-induced PpIX in P3 spheroids after the radiotherapy, please indicate results. Moreover, please discuss acuity of porphyrin synthesis in cancer cell (cancer stem-cell) after the radiotherapy.

2)Because 5-ALA is not porphyrin, function of 5-ALA is multifactorial and complex for human body. Many studies indicated immunological effect of 5-ALA in photodynamic therapy for cancer cells. I wonder repeated administration of 5-ALA affect immunological system during the RDT.  If possible, please discuss that point.

3)In this radiation protocol of animal study, “5 times to the combination treatment 5-ALA/radiotherapy (10 Gy)” means fractionated radiotherapy in 5 consecutive days with 5-ALA administration 4 hours before each irradiation (on day 39-43)?  Similarly, were 5-ALA administered in 5 consecutive days in control group (Figure 3)? Are there control group without 5-ALA administration? Please indicate precise protocol, if possible, using schema. I think it is very important point.

4)If you perform pathological examination after the brain tumor model in 5-ALA RDT, please discuss.

Author Response

Response to Reviewer 1 Comments

The authors investigated radiodynamic effect of 5-aminlevulinic acid (ALA) in brain tumor model which inoculated P3 cell (human stem-like glioblastoma) into the brain of immunodeficient mice. Although 5-ALA can accumulate much amount of Protoporphyrin IX(PpIX) in P3-derived spheroids, radiodynamic effect of 5-ALA was not observed in both P3-derived spheroids and orthotopic model. This study is well designed, and interesting. However, some questions should be raised.

We would like to thank referee for useful comments and constructive criticism of our work. We hope that our taking into account suggestions will bring our study eligible for publication in Cancers.

Major concern

Point 1: Radiosensitizing effect of porphyrin compounds have been observed in cancer cells. Because 5-ALA is prodrug of heme, amount of 5-ALA-induced porphyrin compounds (i.e. PpIX) is low in cancer cells, compared to other porphyrin, such as HpD and Photofrin. Therefore, radiodynamic effect of 5-ALA is considered as low in cancer cells via the single dose radiation. However, acuity of porphyrin synthesis has changed in cancer cells after radiotherapy with some variation. If you confirm accumulation of 5-ALA-induced PpIX in P3 spheroids after the radiotherapy, please indicate results. Moreover, please discuss acuity of porphyrin synthesis in cancer cell (cancer stem-cell) after the radiotherapy.

Response 1: We agree with the reviewer that exogenous mixes of porphyrins, especially if they are not metabolized by the heme pathway, may lead to higher intracellular accumulation of porphyrins than 5-ALA treatment, and we did mention this point in the introduction (reference 17 in our manuscript).This point could be nuanced by  Schaffer and al. (1) who showed that the intratumoral amounts of porphyrins in subcutaneously transplanted bladder carcinoma nude mice were almost the same after 5-ALA treatment at 200mg/kg (0.97µg/g) compared to treatment with Photofrin (15mg/kg and 1.03µg/g of porphyrins) and greater than treatment with haematoporphyrin (10mg/kg and 0.57µg/g of porphyrins).

Multifractionation for radiodynamic effect enhancement of porphyrin accumulation after 5-ALA administration has been well illustrated by Yamamoto et al.(2) who showed that multifractionation was effective in vitro and in vivo to radiosensitize with 5-ALA in 2 primary tumour cell lines(3). To answer the reviewer’s comment on the impact of radiotherapy on porphyrin, we conducted an experiment measuring the accumulation of porphyrins after repeated overloads of 5-ALA post-radiotherapy, (figure 1 below), as compared to repeated exposures to 5-ALA without radiotherapy.

Figure 1: Radiotherapy promotes 5-ALA-induced porphyrin accumulation on the long term. Porphyrin accumulation was measured every 24h by flow cytometry after multiple 5-ALA exposures of P3-spheroids for 4h, with or without a single 4Gy irradiation.

We show that there is a 5-ALA-derived accumulation of porphyrins in spheroids even after a single irradiation. The amount of porphyrins seems also greater and greater with repeated 5-ALA exposures (58105<67598<83151 MFI, versus around 60000 at each control non-irradiated point). This is not due to lack of porphyrin clearance since 24 hours after 5-ALA exposure, the amounts of porphyrins are strongly decreased (3350 or 2428), in agreement with figure 5B of the manuscript. Thus, as suggested by the reviewer, lasting porphyrin accumulation seems to be reinforced after radiotherapy, which would promote a potential radiodynamic effect. Accordingly, our experimental in vivo protocol included 5-ALA was administration each time the animals were treated by radiotherapy.

Point 2: Because 5-ALA is not porphyrin, function of 5-ALA is multifactorial and complex for human body. Many studies indicated immunological effect of 5-ALA in photodynamic therapy for cancer cells. I wonder repeated administration of 5-ALA affect immunological system during the RDT.  If possible, please discuss that point.

Response 2: We used a PDX model where human-derived tumour cells with stem cell features were xenografted in immune-deficient mice. So we did not explore this point. However, we agree with the reviewer that when using an immune-competent model, the immune system might participate in the antitumour effect. We discussed this point in the discussion as follows: “A phototherapeutic effect in subcutaneous tumours cannot be excluded (20). In addition, this immune-competent model is interesting because the authors found increased macrophage tumour infusion in the 5-ALA treated rats probably contributing to the tumoricidal effect ».

Point 3: In this radiation protocol of animal study, “5 times to the combination treatment 5-ALA/radiotherapy (10 Gy)” means fractionated radiotherapy in 5 consecutive days with 5-ALA administration 4 hours before each irradiation (on day 39-43)?  Similarly, were 5-ALA administered in 5 consecutive days in control group (Figure 3)? Are there control group without 5-ALA administration? Please indicate precise protocol, if possible, using schema. I think it is very important point.

Response 3: Radiotherapy was performed 3 times a week on Monday, Wednesday and Friday. In the RT+5-ALA group, 5-ALA was administrated each time the animals were treated by radiotherapy. This sentence has been added in the material and methods section.

There were no control group without 5-ALA administration as mentioned in the manuscript (lines 223-225) « To test the radiosensitization potential of porphyrin accumulation, 3 groups of 11 mice were used. One group received only 5-ALA, one group received radiotherapy alone and the third group received both. »

We now provide a scheme to clarify the radiotherapy protocole at a glance and inserted it in figure 3.

Figure 3. Combination treatment of 5-ALA and radiotherapy.

A : experimental protocol (details in materials and methods).Black arrows indicate 5-ALA administration 4h before irradiation. Each star correspond to luminescence imaging.
/f: per fraction ; 3d/w: 3 days a week ; IP: intraperitoneal. Mouse picture provided by Servier Medical Art

Point 4: If you perform pathological examination after the brain tumor model in 5-ALA RDT, please discuss.

Response 4: We did not perform pathological examination of the brain after 5-ALA+RT treatment as there were no difference with RT only in term on tumour growth or mouse survival.

Bibliography

  1. Schaffer M, Schaffer PM, Corti L, Gardiman M, Sotti G, Hofstetter A, et al. Photofrin as a specific radiosensitizing agent for tumors: studies in comparison to other porphyrins, in an experimental in vivo model. J Photochem Photobiol B. 2002 Apr;66(3):157–64.
  2. Yamamoto J. Radiosensitizing effect of 5-aminolevulinic acid-induced protoporphyrin IX in glioma cells in�vitro. Oncol Rep [Internet]. 2012 Feb 28 [cited 2022 Apr 13]; Available from: http://www.spandidos-publications.com/10.3892/or.2012.1699
  3. Yamamoto J, Ogura S-I, Shimajiri S, Nakano Y, Akiba D, Kitagawa T, et al. 5-aminolevulinic acid-induced protoporphyrin IX with multi-dose ionizing irradiation enhances host antitumor response and strongly inhibits tumor growth in experimental glioma in vivo. Mol Med Rep. 2015 Mar;11(3):1813–9.

Reviewer 2 Report

The study investigates the hypothesized radiosensitization that the 5-ALA marker would induce on the GB.

The topic is moderately interesting, however it clarifies in general very little some aspects.

In the summary "

Radiosensitization of glioblastoma is a major ambition to increase the survival 17 of this incurable cancer. Before surgery, oral administration of 5 aminolevulinic acid leads to the 18 accumulation of fluorescent protoporphyrin IX, preferentially in glioblastoma as compared to nor- 19 mal brain. This property is used to optimize the resection of the tumour under fluorescent light. 20 Protoporphyrin IX may also carry radiosensitization activity" is not complete and not clear enough.

The study is well conducted, but poorly and poorly introduced. The 5-ALA is a fluorescence marker used in the resection of gliomas that should provide aid to achieving the correct extent of resection that represents the FIRST STEP fundamental to ensure the best prognosis for the patient, this concept is not included in the initial part of the paper.

"P3 cells, human stem-like glioblastoma cells (21) were transduced by 87 the lentiviral virus coding for Luciferase" this is not the only type of tumor-growth and method achievable  for these type of study. The authors need to explain in a further section, if the study in their opinion may give the same results in other cultures of GB cells, perhaps at different radiosensitivity, to ascertain that there is no effect.

In discussion section:

"our irradiation setting was as close as possible 327 to what is routinely used in patients and was not toxic for the mice. Of note, we used 328luciferase/luciferin for the tumour follow-up. Luciferase photon emission could interfere 329 with a phototherapeutic effect. However, the administration of 5-ALA and luciferin were 330 always distant by 24h, and we showed that PpIX was cleared during this period." In my opinion, this is no small limitation.

General grammar revision

Recommend major revision

Author Response

Response to Reviewer 2 Comments

The study investigates the hypothesized radiosensitization that the 5-ALA marker would induce on the GB.

The topic is moderately interesting, however it clarifies in general very little some aspects.

We would like to thank referee for useful comments and constructive criticism of our work. We hope that our taking into account suggestions will bring our study eligible for publication in Cancers.

Point 1: In the summary "Radiosensitization of glioblastoma is a major ambition to increase the survival 17 of this incurable cancer. Before surgery, oral administration of 5 aminolevulinic acid leads to the 18 accumulation of fluorescent protoporphyrin IX, preferentially in glioblastoma as compared to nor- 19 mal brain. This property is used to optimize the resection of the tumour under fluorescent light. 20 Protoporphyrin IX may also carry radiosensitization activity" is not complete and not clear enough.

Response 1: We agree with the reviewer, and propose to amend the Simple Summary this way :

Radiosensitization of glioblastoma is a major ambition to increase the survival of this incurable cancer. A first step to ensure improve the prognosis is to achieve the best extent of resection. Oral administration of 5 aminolevulinic acid before surgery leads to the accumulation of fluorescent protoporphyrin IX in glioblastoma as compared to normal brain. Fluorescence-guided surgery thanks to protoporphyrin IX increases complete resection and disease free survival. Protoporphyrin IX may also carry radiosensitization property.

Point 2: The study is well conducted, but poorly and poorly introduced. The 5-ALA is a fluorescence marker used in the resection of gliomas that should provide aid to achieving the correct extent of resection that represents the FIRST STEP fundamental to ensure the best prognosis for the patient, this concept is not included in the initial part of the paper.

Response 2: We agree with the reviewer, and propose to add in the introduction :

“This property was exploited in a subsequent randomized phase III clinical trial to compare classical neurosurgery to fluorescence-guided surgery after 5-ALA administration. PpIX-guided tumour visualization increased complete resection from 36% to 65%. Progression-free survival was better, however, at the cost of neurological deficits, and this strategy did not improve overall survival (10). However, Eatz et al. (1), performed a recent review on 45 studies using fluorescence-guided vs white light surgery. 5-ALA administration improved overall survival and disease free survival. These favorable outcomes were associated with correct extend of resection, the first step to ensure best prognosis. Eatz et al. also observed discripancies between studies when considering post-surgery neurological deficits, making this point still inconclusive.”

Point 3: "P3 cells, human stem-like glioblastoma cells (21) were transduced by 87 the lentiviral virus coding for Luciferase" this is not the only type of tumor-growth and method achievable  for these type of study. The authors need to explain in a further section, if the study in their opinion may give the same results in other cultures of GB cells, perhaps at different radiosensitivity, to ascertain that there is no effect.

Response 3: To our knowledge, there are 2 methods to follow intracranial tumour-growth live, which are radiology (best by MRI) and Bioluminescence. MRI was not chosen because of the number of mice per evaluation (33), and number of evaluations (1/w for 11-12 weeks). MRI at each time point would take around 15-30 minutes per animal, and between 8h and 16h for all 3 groups. Bioluminescence was more adapted to this sample size. Moreover, a good correlation was found between bioluminescence and MRI for intra-cranial tumour follow-up (2).

The second point of the reviewer concerns the response of distinct GB cell lines to our experimental protocol. We fully agree with the reviewer and did mention this in the discussion section: “As heterogeneity between tumour cells seems to impact response to 5-ALA treatment, other glioblastoma PDXs might be sensitive to combination treatment. This hypothesis deserves further exploration.” We did add comments on the initial radiosensitivity of the tumours as follows: “In our opinion, different results could be found according to glioblastoma radiosensitivity.”

Point 4: In discussion section:
"our irradiation setting was as close as possible 327 to what is routinely used in patients and was not toxic for the mice. Of note, we used 328luciferase/luciferin for the tumour follow-up. Luciferase photon emission could interfere 329 with a phototherapeutic effect. However, the administration of 5-ALA and luciferin were 330 always distant by 24h, and we showed that PpIX was cleared during this period." In my opinion, this is no small limitation.

Response 4: After Luciferin injection, the maximal luminescence plateaus after around 10 minutes, depending on the tumour size. According to our experience and published data, the maximum signal lasts for less than an hour and the luciferin substrate is mostly cleared from the tumour in less than 2h(3). Thus, when mice are exposed to 5-ALA and produce porphyrins in their tumours, luciferin substrate is cleared from the animals and the tumour bioluminescence is null. Conversely, when mice are injected with luciferin substrate, porphyrin accumulation has been resolved by the heme pathway (Figure 2). However, we understand that our experimental protocol was not clear enough. We now provide details in the M&M section and in Figure 3A.

Figure 3. Combination treatment of 5-ALA and radiotherapy. A. experimental protocol (details in materials and methods).Black arrows indicate 5-ALA administration 4h before irradiation. Each star correspond to luminescence imaging. B. Tumour follow-up by bioluminescence (Mean +/-SEM). N=11/group. Arrows indicate the days after implantation where mice received fractionated radiotherapy 5 times 2Gy +/- 5-ALA. The control group received 5-ALA only. BC. Corresponding survival curves.

/f: per fraction ; 3d/w: 3 days a week ; IP: intraperitoneal. Mouse picture provided by Servier Medical Art

Bibliographie

  1. Eatz TA, Eichberg DG, Lu VM, Di L, Komotar RJ, Ivan ME. Intraoperative 5-ALA fluorescence-guided resection of high-grade glioma leads to greater extent of resection with better outcomes: a systematic review. J Neurooncol. 2022 Jan 1;156(2):233–56.
  2. Jost SC, Collins L, Travers S, Piwnica-Worms D, Garbow JR. Measuring Brain Tumor Growth: Combined Bioluminescence Imaging–Magnetic Resonance Imaging Strategy. Mol Imaging. 2009 Sep 1;8(5):7290.2009.00023.

3.            Paroo Z, Bollinger RA, Braasch DA, Richer E, Corey DR, Antich PP, et al. Validating Bioluminescence Imaging as a High-Throughput, Quantitative Modality for Assessing Tumor Burden. 2004;3(2):8. 

Round 2

Reviewer 1 Report

The authors clearly answered my comments, and this revised version of the manuscript has been improved.

Reviewer 2 Report

The structure of the paper in the abstract and introduction now has a more interesting design, the study is useful to substantiate a technique of using 5-ALA that in my opinion has the sole purpose of improving surgical resection. I only recommend not to use the acronym GBM and replace it with GB by virtue of the latest WHO 2021 classification that omits the appellation "multiform". 
I also recommend to report some clinical studies that further clarify the usefulness of the use of 5-ALA in surgery: Armocida D, Pesce A, Di Giammarco F, Frati A, Salvati M, Santoro A. Histological, molecular, clinical and outcomes characteristics of Multiple Lesion Glioblastoma. A retrospective monocentric study and review of literature. Neurocirugia (Astur : Engl Ed). 2021 May-Jun;32(3):114-123. English, Spanish. doi: 10.1016/j.neucir.2020.04.003. Epub 2020 Jun 18. PMID: 32564972.